



# Case study on the influence of synoptic-scale processes on the paired H₂O–O₃ distribution in the UTLS across a North Atlantic jet stream

Andreas Schäfler[1], Michael Sprenger[2], Heini Wernli[2], Andreas Fix[1], Martin Wirth[1]

[1]Deutsches Zentrum für Luft- und Raumfahrt, Institut für Physik der Atmosphäre, Oberpfaffenhofen, Germany

[2]Institute for Atmospheric and Climate Science, ETH Zurich, Switzerland

*Correspondence to*: Andreas Schäfler (Andreas.Schaefler@dlr.de)

**Abstract.** During a research flight of the Wave-driven ISentropic Exchange (WISE) campaign, which was conducted over the eastern North Atlantic on 1 October 2017, the composition of the Upper Troposphere and Lower Stratosphere (UTLS) across the North Atlantic jet stream was observed by airborne, range-resolved Differential Absorption Lidar (DIAL) profiles. We

investigate how the high variability in the paired H₂O and O₃ distribution along the two-dimensional lidar cross section is affected by synoptic-scale weather systems, as revealed by the Lagrangian history of the observed air masses. To this aim, the lidar observations are combined with 10-day backward trajectories along which meteorological parameters and derived turbulence diagnostics are traced. The transport and mixing characteristics are then projected to the vertical cross sections of the lidar measurements and to the H₂O–O₃ phase space to explore linkages with the evolution of synoptic scale weather systems

and their interaction. Tropical, midlatitude and arctic weather systems in the region of the jet stream and the related transport and mixing explain the complex H₂O and O₃ distribution to a large extent: O₃-rich stratospheric air from the high Arctic interacts with midlatitude air from the North Pacific in a northward deflected jet stream associated with an anticyclone over the US and forms a filament extending into the tropopause fold beneath the jet stream. In the troposphere, lifting related to convection in the innertropical convergence zone (ITCZ) and two tropical cyclones continuously injected H₂O into dry

descending air from the tropical Atlantic and Pacific forming filamentary H₂O structures. One tropical cyclone that transitioned into a midlatitude cyclone lifted moist boundary layer air explaining the highest tropospheric H₂O values. During the two days before the observations the air with mixed tropospheric and stratospheric characteristics experienced frequent turbulence along the North Atlantic jet stream indicating a strong influence of turbulence on the formation of the Extratropical Transition Layer (ExTL). This investigation highlights the complexity of stirring and mixing processes and their close connection to interacting

tropospheric weather systems from the tropics to polar regions, which strongly influenced the observed fine-scale H₂O and O₃ distributions. The identified non-local character of mixing should be kept in mind when interpreting mixing lines in tracer–tracer phase space diagrams.



## 1 Introduction

The extratropical upper troposphere and lower stratosphere (UTLS) is the transition region between troposphere and
stratosphere that exhibits rapid changes in thermodynamic quantities and chemical constituents (Gettelman et al., 2011). The
UTLS composition around the tropopause is altered by two-way exchange processes between both layers (Stohl et al., 2003;
Holton et al., 1995). While fast mixing time scales in the troposphere quickly fade out the impact of the stratosphere (Shepherd,
2007), persistent intermediate chemical characteristics are found within the so-called extratropical transition layer (ExTL) in
the LS (Hoor et al., 2004; Pan et al., 2004; Hegglin et al., 2009). The distribution of radiatively active trace gases in the UTLS
is of key importance for the Earth's radiation budget, and thus knowledge about processes that impact their distribution is
relevant for accurately predicting weather (Chagnon et al., 2013; Shepherd et al., 2018; Bland et al., 2021) and climate (Riese
et al., 2012).

A variety of dynamical processes, which act on different temporal and spatial scales and interact with the chemistry, influence
the composition of the midlatitude UTLS and the ExTL therein (Gettelman et al., 2011). On longer time scales (months to
years) downwelling within the Brewer-Dobson circulation transports chemical species downward into the extratropical LS
(Shepherd, 2007; Birner and Bönisch, 2011). On synoptic time scales (up to 10 days) the continuous sequence of propagating,
amplifying and eventually breaking Rossby waves and related baroclinic cyclones redistribute air in the midlatitudes and
impact trace gas distributions through stirring (Appenzeller et al., 1996) and mixing (Škerlak et al., 2014; Boothe and Homeyer,
2017) of neighbouring tropospheric and stratospheric air masses. On even shorter mesoscale time scales deep convection can
modify the UTLS composition by rapid upward transport of boundary layer air and injection into the LS (Hegglin et al., 2004;
Homeyer et al., 2014).

A central approach to interpret trace gas distributions and to understand the transport history and exchange of air across the
tropopause is based on Lagrangian modelling, which profited from continuously improved spatial and temporal resolution of
the driving wind fields (typically from reanalyses) as well as of the modelling of chemistry along the pathways. Konopka and
Pan (2012) found that (i) mixing processes on a time-scale of three days are crucial for the formation of the ExTL, (ii) very
distinct air masses are involved in the transport, and (iii) mixing is largely controlled by the upstream synoptic situation.
Distinct transport pathways from different source regions can contribute to the mixed air in the ExTL surrounding the
midlatitude jet stream (Vogel et al., 2011). Related Rossby wave breaking (RWB) events, which are associated with stirring
and mixing of air masses, can cause small-scale trace gas filaments (Ungermann et al., 2013; Krasauskas et al., 2021). RWB-
induced poleward extrusions of tropospheric air also affect the ExTL composition (Pan et al., 2009). In summary, numerous
studies applied Lagrangian modelling and highlight that dynamical processes on synoptic time scales strongly impact the
formation of the ExTL (e.g., Pan et al., 2007; Homeyer et al., 2011).

Mixing and transport processes in association with Rossby waves over the North Atlantic were studied in autumn 2017 during
the Wave-driven ISentropic Exchange (WISE) campaign (Kunkel et al., 2019). In the course of WISE, the chemical
discontinuity at the extratropical tropopause could be characterized for the first time by collocated lidar profile observations,



which provided a two-dimensional view on the paired $H_2O$ and $O_3$ distribution in the UTLS. The present study builds upon a first study analysing the observations during the flight on 1 October 2017 (Schäfler et al., 2021), which highlighted the following key findings and open issues:

- $O_3$ and $H_2O$ observations along a meridional cross section (Fig. 1a and b) intersecting the North Atlantic jet stream (Fig. 1c) were obtained in a region where isentropes intersect with the tropopause ["middleworld" following the convention of Hoskins (1991)], i.e., a region of frequent isentropic mixing (Holton et al., 1995; Stohl et al., 2003). It was hypothesised that the tropospheric $H_2O$ variability (Fig. 1a) is associated with differing tropospheric transport pathways of originally tropical/subtropical and extratropical air. In the poleward located LS, filamentary structures of increased $O_3$ mixing ratios extend into a tropopause fold beneath the jet stream (Fig. 1b). Although, it is known that the ageostrophic circulation (Keyser and Shapiro, 1986) and isentropic differential advection (Hitchman and Rowe, 2021) contribute to tropopause folding (Danielsen et al., 1987) and filamentation of air masses, the formation mechanisms of the cross-isentropic stratospheric filaments within this particular fold remained open.

- The small-scale spatial variability in the trace gas distributions contrasts the compact distribution of tropospheric, stratospheric and mixed (ExTL) air (Fig. 1f) that were determined by frequency distributions in tracer–tracer (T–T) space (Fig. 1d and 1e). Unmixed background air exhibits an L-shaped distribution with a distinct and almost linear tropospheric (low $O_3$, variable $H_2O$) and stratospheric branch (variable $O_3$, low $H_2O$). In contrast, mixed air was defined by its intermediate $O_3$ and $H_2O$ concentrations connecting troposphere and stratosphere along mixing lines. The rapid upward decrease of tropospheric $H_2O$ (Fig. 1a) led to a characteristic clustering of mixing lines (Fig. 1d) and thus to a subdivision of the mixed and tropospheric air mass classes (Fig. 1e and f). In the upper part, stratospheric air (STRA) is connected with dry tropospheric air (TRO-1/TRO-2) through MIX-1 while below the jet maximum MIX-2 mixing lines couple with moist midlatitude air (TRO-3). It was hypothesized that different transport pathways brought air of different $H_2O$ volume mixing ratio (VMR) to the upper troposphere. Dry tropospheric air (TRO-1 and TRO-2) was potentially either dehydrated in tropical convection or lifted in a tropical cyclone (TC). The extended moist air layer (TRO-3a) was supposed to be related to more recent ascents from the midlatitude planetary boundary layer (PBL).

- It was argued that the chemical composition in the ExTL is rather an effect of mixing events along the history of the air masses than quasi-instantaneous mixing at the time of the observation. Differences in the homogeneity along the mixing lines in T–T space were found for individual regions along the observed cross section. More uniform transitions occurred in the lower tropopause fold and the upper part of the jet stream, which may indicate recent mixing events and a crucial role of the dynamic situation on the observed $O_3$ and $H_2O$ concentrations. We hypothesized that non-local clear air turbulence (CAT) related to the strong wind shear in the upstream jet stream (Shapiro, 1980; Spreitzer et al., 2019) and upper- and lower-level frontal zones (Keyser and Shapiro, 1986; Lang and Martin, 2012) may have fostered the mixing in this case.

These findings agree with the above-mentioned studies on the importance of transport and mixing processes, but they raise several follow-on questions about the formation of the highly variable $H_2O$ and $O_3$ distributions. In this study we investigate



to what extent the observed fine-scale composition of the UTLS is related to tropospheric weather systems, which occur on synoptic time scales (up to 10 days) and can substantially impact transport and mixing. We expand on the analysis of the role of individual synoptic-scale weather systems and also study their interplay. We suppose this interplay to be of crucial relevance for better understanding $H_2O$ and $O_3$ distributions in the UTLS, as portrayed by the collocated airborne DIAL observations on this particular jet stream crossing. In addition, the temporal and spatial distribution of turbulent mixing is investigated to better

understand the formation of mixing lines in the ExTL. The observations along the lidar cross section are combined with 10-day air parcel trajectories that allow meteorological quantities and specific turbulence indicators to be traced backward in time. The derived characteristics are then projected to the vertical cross sections and to the $H_2O$–$O_3$ phase space to analyse transport and mixing in connection with the evolution of synoptic scale weather systems.

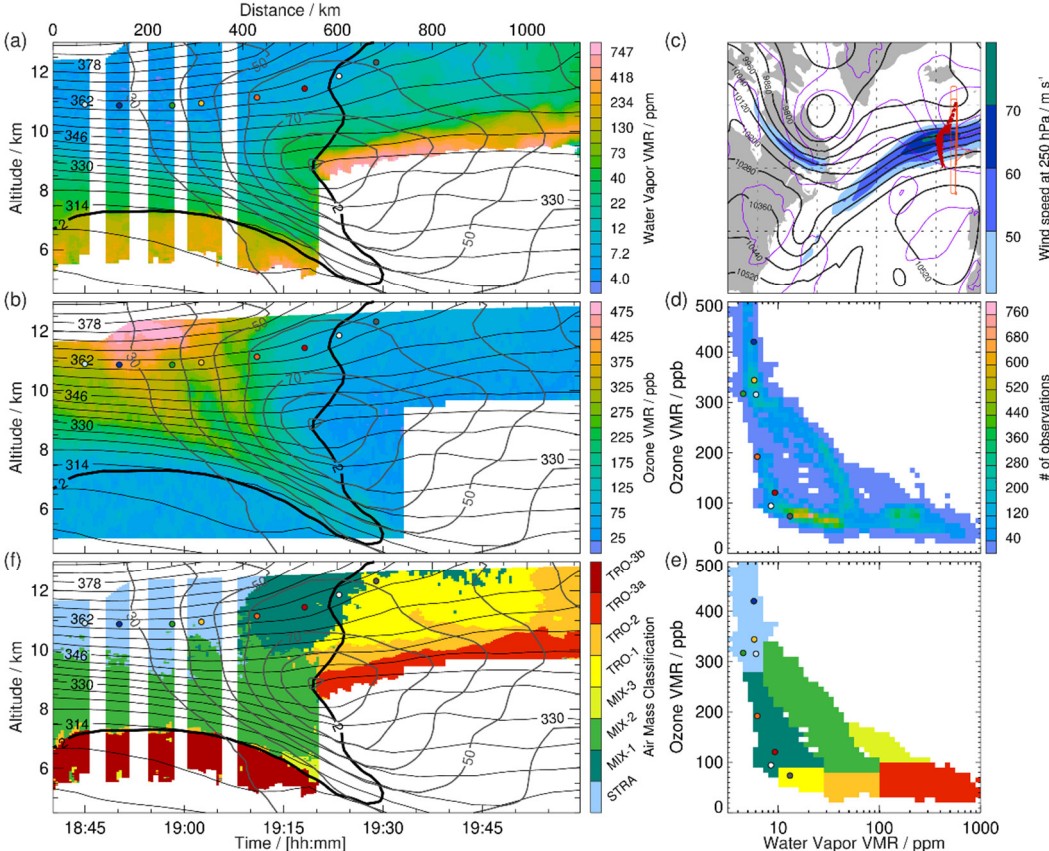

**Figure 1:** (a) DIAL $H_2O$ volume mixing ratio (VMR; in ppm = µmol mol$^{-1}$) and (b) $O_3$ VMR (in ppb = nmol mol$^{-1}$) observed on 1 October 2017. (c) Geopotential height (thick black contours) and wind speed (colour shading) at 250 hPa as well as mean sea level pressure (purple contours) at 18:00 UTC superimposed by the flight track (red line with bold north to south flight leg considered in this paper). Thick dark red dots indicate the location of the observed air at 18:00 UTC (for details see text). (d) Frequency distributions of collocated observations in tracer–tracer (T–T) space. (e) Air mass classification in T–T space and (f) its reprojection to geometrical space. Please note that the

TRO-3 air mass in (f) is separated in a part south (TRO-3a) and north (TRO-3b) of the jet stream. Panels a, b and f are superimposed by horizontal wind speed (grey contours; in m s$^{-1}$ for wind speeds > 30 m s$^{-1}$), potential temperature (black contours; in K) and dynamical tropopause (2 potential vorticity units (PVU); 1 PVU = 10$^{-6}$ K kg$^{-1}$ m$^2$ s$^{-1}$; thick black contour). Coloured dots mark starting locations of trajectories shown in Fig. 7. Please note that a separation of TRO-3a and TRO-3b is only possible in the cross section (f) and that the air overlaps in (e). Figure based on Schäfler et al. (2021).





In Sect. 2 the data and methods are described. Section 3.1 investigates the transport pathways for the individual air mass classes, while in Sect. 3.2 the role of tropospheric and stratospheric transport is discussed. Section 3.3 connects the transport pathways with the dominant weather systems. Section 3.4 investigates specific aspects related to tropical and Arctic transport. In Sect. 3.5 the role of mixing processes and their relation to individual weather systems are addressed. A discussion of the results is given in Sect. 4 before Sect. 5 summarizes and concludes this work.

## 120  2 Data and methods

### 2.1 Observations

During WISE, the four-wavelength Differential Absorption Lidar (DIAL) WAter vapor differential absorption Lidar Experiment in Space (WALES) was measuring $H_2O$ at 935 nm and $O_3$ at 305/315 nm making use of the differential absorption technique (Wirth et al., 2009; Fix et al., 2019). It was the first deployment of this setup to observe the UTLS in a nadir-pointing

mode onboard the HALO aircraft (Krautstrunk and Giez, 2012). This study is based on WISE measurements performed on 1 October 2017 between 18:40 and 20:00 UTC (see Fig. 1), which have a 24 s temporal resolution corresponding to 5.6 km profile distance in the horizontal. Vertically, $H_2O$ and $O_3$ are determined every 15 m, although the effective vertical resolution is approximately 500 m (full width at half maximum, FWHM, of the averaging kernel). The observed number density from the DIAL is converted to VMR. In total 57135 collocated observations are considered. The data products and the combined

analysis method of the collocated $H_2O$ and $O_3$ lidar observations in T–T space to characterize the UTLS (as introduced in Sect. 1) are described in greater detail in Schäfler et al. (2021).

### 2.2 Model data and trajectory calculations

Observational data is supplemented by fields from the European Centre for Medium-Range Weather Forecasts (ECMWF) ERA5 reanalysis (Hersbach et al., 2020) available at a one-hourly temporal resolution and interpolated on a 0.5°×0.5°

latitude/longitude grid with 137 vertical levels. These reanalysis fields were interpolated bilinearly in space and linearly in time towards the observation location (Schäfler et al., 2010).

Three-dimensional one-hourly ERA5 model level winds were also used to drive kinematic offline trajectory calculations that were performed with the Lagrangian Analysis Tool (LAGRANTO, Wernli and Davies, 1997; Sprenger and Wernli, 2015) for all 57135 observation locations. To account for the temporal offset between the 200 lidar profiles, the starting time of the

trajectories was varied along the lidar section in five-minute intervals. Hence, all lidar profiles within five minutes received the same trajectory starting time while their spatial location, defined by latitude, longitude and pressure, is exact. Trajectories were first calculated backward with a five-minute output until 18:00 UTC on 1 October 2017 and then extended over 10 days until 18:00 UTC on 21 September 2017 with one-hourly output. At some places we refer to the origin of air masses that we define as the location at the earliest time. Beside the trajectory position, several physical parameters were traced over the ten-

day period including temperature, potential temperature and potential vorticity.





In addition, turbulence indicators were calculated on ERA5 model levels and subsequently traced along the trajectories. Such turbulence diagnostics assume that favorable environments for turbulence can be predicted from coarser NWP output assuming that energy is transferred from large, resolved NWP scales down to small-scale turbulence (Sharman et al., 2012; Sharman, 2016; Storer et al., 2019). Different diagnostics, providing individual advantages and disadvantages depending on the

dynamical situation were applied that are often jointly considered in turbulence forecasting (Ellrod et al., 2003; Sharman et al., 2006). Here we considered two conventional parameters, the Richardson number (Ri) and the semi-empirical Ellrod-Knapp turbulence index (TI). Both indicators are useful to predict wind shear-induced turbulence related to jet streams. It has to be noted that Ri and TI do not forecast convective or mountain-wave turbulence, which are other potentially important sources of turbulent instability (Storer et al., 2019). Ri and TI were successfully applied to study stratosphere–troposphere exchange

(STE) and its link to turbulence processes (Traub and Lelieveld, 2003; Karpechko et al., 2007; Kunkel et al., 2019). First, Ri is defined as the ratio of the squared Brunt-Vaisala frequency and squared vertical wind shear (Stull, 1988):

$$Ri = \frac{\frac{g}{\theta_v} \frac{\partial \theta_v}{\partial z}}{\left(\frac{\partial u}{\partial z}\right)^2 + \left(\frac{\partial v}{\partial z}\right)^2},$$

where $\theta_v$ is the virtual potential temperature, $u$ and $v$ are the horizontal wind components and $g$ is the acceleration due to gravity. Ri quantifies the inhibition of turbulence due to stable stratification vs. mechanically wind shear-driven turbulence

production. Small Ri values (in theory smaller than 0.25), related to high shear and unstable environments, are indicator for Kelvin-Helmholtz instability (Stull, 1988), which is the key process leading to CAT and mixing of neighboring air masses. CAT can modify local gradients of winds (wind shear), temperature (stability) but also of trace gases (Kunkel et al., 2019). As Ri values calculated for finite layers from observational or gridded NWP data cannot resolve the small theoretical Ri values (Storer et al., 2019) a threshold of Ri < 2 is used here as a criterion for regions where perturbations due to shear instabilities

can grow and eventually develop into turbulence.

Second, we calculate the TI-Index that combines vertical wind shear with horizontal deformation (Ellrod and Knapp, 1992) as follows:

$$TI = \left(\left(\frac{\partial u}{\partial z}\right)^2 + \left(\frac{\partial u}{\partial z}\right)^2\right)^{1/2} \left(\left(\frac{\partial u}{\partial x} - \frac{\partial v}{\partial y}\right)^2 + \left(\frac{\partial v}{\partial x} + \frac{\partial u}{\partial y}\right)^2\right)^{1/2}$$

TI is suitable to detect CAT in upper-level jet streams and frontal zones as it accounts for horizontal deformation of the flow

that can locally trigger turbulence or modify temperature and related wind gradients (Storer et al., 2019). The probability of CAT increases with increasing TI values. Typically used thresholds of TI in the literature range between 2–12 x $10^{-7}$ s$^{-1}$ (Ellrod and Knapp, 1992; Jaeger and Sprenger, 2007). In this study TI > 8 x $10^{-7}$ s$^{-1}$ is applied.

For both Ri and TI, only events at a pressure smaller than 500 hPa are considered in order to exclude lower tropospheric turbulent processes that are not of interest to this work. The climatological study by Jaeger and Sprenger (2007) showed

significant differences for the distributions of Ri and TI relative to the jet stream and therefore it is reasonable to consider them in a complementary way.


## 3 Results

### 3.1 Pathways related to air masses classes

To ease the following discussion about transport pathways, we first summarize the distribution of air mass classes along the
lidar cross section following Schäfler et al. (2021). TRO-1 and TRO-2 occur to the south of the jet stream at high altitudes. Unlike shown in Schäfler et al. (2021), TRO-3 is separated into a region south (TRO-3a) and north (TRO-3b) of the jet stream (Fig. 1f). Stratospheric background air (STRA) occurs at the highest altitudes north of the jet stream. In between, the ExTL is located that is composed of the tropopause following MIX classes. These three mixing regimes, which are defined in T–T space (Fig. 1d), connect STRA with different parts of the troposphere. MIX-1 occurs at high altitudes to the south of STRA,
MIX-2 below MIX-1 and STRA, and MIX-3 as a small region below MIX-2 (not considered in Fig. 2).

First, gridded trajectory locations are visualized in Fig. 2 for these individual air mass classes along the flight track to give an overview on the transport pathways. Note that high values in Fig. 2 may be related to slow and/or coherent transport. STRA air (Fig. 2a) remains in the Artic during the 10 days before the observations, with a coherent transport over the North Atlantic, Labrador and Baffin Island, and more widespread transport upstream over the high Arctic and eastern Russia. MIX-1
trajectories (Fig. 2b) are associated with relatively coherent meandering midlatitude transport over the eastern North Pacific and across the US and North Atlantic. Further upstream, MIX-1 air partly originates in the subtropics but also exhibits fast zonal transport from the western North Pacific, China and even northern Africa. The transport of MIX-2 (Fig. 2c) is comparable to MIX-1 over the Atlantic, but upstream the flow is less coherent with transport over northern Canada, the northern North Pacific, eastern Russia and China. TRO-1 and TRO-2 (Fig. 2d and e) have very similar transport pathways from the subtropical
and tropical eastern North Pacific over Mexico and southeastern US towards the flight track, which indicates that the separation to distinguish the tropospheric end of MIX-1 and MIX-2 (see Fig. 1e) does not correspond to different pathways. Finally, TRO-3 (Fig. 2f) splits into two distinct branches that are related to TRO-3a and TRO-3b (Fig. 1f). The air in the northern part (TRO-3b) approaches the flight track from the west with a zonal transport over Canada, while the southern part (TRO-3a) has its origin over the midlatitude North Atlantic, which is clearly different to TRO-1 and TRO-2.
In summary, all classes are characterized by a comparatively coherent transport over the North Atlantic but regarding the origins and pathways, they differ substantially. This is remarkable given that observations were taken over a relatively short flight distance of ~1100 km (50.5° to 60.5° N).



**Figure 2:** Gridded number of trajectory positions for air mass classes divided by the total number of trajectories per class (as in Fig. 1) in 2.5°x2.5° boxes for the 10-day period from 1 October to 22 September 2017. Grey circles mark the Tropic of Cancer and the Arctic Circle.

## 3.2 Tropospheric and stratospheric transport

In a next step the 10-day transport is discussed separately for the troposphere and the lower stratosphere. For this purpose, information about traced variables along the individual backward trajectories is projected onto the lidar curtain as well as into T–T space. Liniger and Davies (2003) referred to this type of visualization, where information along the transport is projected to the trajectories' starting points, as Lagrangian forward projections.

First, we consider tropospheric air (PV < 2) south and north of the jet stream. Figures 3a and b show that TRO-1/TRO-2 are originating most southerly (min. latitudes < 23° N) indicating transport from the tropics while for TRO-3a the minimum latitude is in the midlatitudes. All tropospheric air reaches maximum latitudes at the time of the observation (Fig. 3c, d), confirming transport from the south (Fig. 2d–f). The part of the tropical air closer to the jet stream is on average located further south (Figs. 3e and f). The high maximum pressure along the 10-day trajectories (Fig. 3g) highlights that midlatitude TRO-3a air originates from the PBL while low maximum pressure for TRO-1/TRO-2, which is reached at the time of observation (not shown), points to a descending transport. However, some embedded filamentary structures of high maximum pressure depict





enhanced vertical transport in TRO-1/TRO-2. In T–T space (Fig. 3h) the moist TRO-3a observations reflect the transport from lowest altitudes while drier TRO-1/TRO-2 air either descends from high altitudes or is lifted before descending. When the observed pressure is compared to the 10-day mean (Fig. 3i, j), TRO-3a shows large values due to the late lifting immediately before the observations. However, the strongest vertical displacement is found for the ascending TRO-1/TRO-2 air parcels that are lifted to highest altitudes at earlier times (not shown). In accordance with weaker lifting, TRO-3a possesses higher minimum temperatures compared to TRO-1 and 2 (Fig. 3k, l), which explains the highest observed $H_2O$ VMRs. TRO-1/TRO-2 air exhibits lowest temperatures at early times and low latitudes and is subsequently warmed (not shown) while descending. Likely these air masses are lifted to the cold tropical tropopause but before the considered 10-day period. Interestingly, the elevated filament with slightly higher $H_2O$ VMRs that is located above the 60 m s$^{-1}$ contour in Fig. 1a originates in the lower troposphere and exhibits slightly warmer minimum temperatures compared to the descending air in its surrounding. The same correlation to a strong ascent is also discernible for increased $H_2O$ VMRs at the end of the flight. TRO-1/TRO-2 ascents appear over the subtropical and tropical Atlantic and partly the western Pacific already 5–10 days before (further details will follow in Sec. 3.3). Thus, fresh low-level air is injected into the northward moving tropical and subtropical air from the western Pacific (see also Fig. 2d and e). TRO-3a lifting occurs over the North Atlantic immediately before the observations. TRO-3b originates in the northern midlatitudes (Fig. 2f and Fig. 3a, e). Note that in T–T space (Fig. 3, right column) TRO-3b is observed in the tropospheric branch with increased $H_2O$ VMR (> 50 ppm) and $O_3$ VMR (> 70 ppb).

Second, stratospheric air (PV > 2) is considered which is characterized by a filamentary region of increased $O_3$ VMR that extends into the tropopause fold (Fig. 1b). This filament features distinct transport characteristics compared to the surrounding air. Its transport reaches the most northern latitudes (Fig. 3c, d) and proceed on average to the north of the rest of STRA and MIX-2 air (Fig. 3e, f). Maximum pressure is reached at the observation location, indicating descending transport which is also reflected in the comparison of observed and 10-day mean pressure (Fig. 3i, j). Apparently, the strongest descent occurs in the tropopause fold which is also obvious in the T–T space showing strongest descent in MIX-2.

In summary, tropospheric transport includes descending air from the tropics, which is partly interrupted by air originating in the PBL leading to increased $H_2O$ VMR at upper levels, and ascending from the midlatitudes. In the stratosphere and ExTL air descends with the northernmost transport being correlated with increased $O_3$ VMR. In the following, we refine this analysis by relating the transport characteristics to tropospheric weather systems.





**Figure 3:** Geometrical (left column) and T–T space (right column) distributions (as in Fig. 1) coloured by characteristics along 10-day backward trajectories: (a, b) minimum latitude, (c, d) maximum latitude, (e, f) difference of observed and 10-day mean latitude, (g, h) mean pressure, (i, j) observed pressure minus 10-day maximum pressure, and (k, l) minimum temperature. Coloured dots mark starting locations of trajectories shown in Fig. 7.





### 3.3 Pathways related to weather systems

The trajectory positions at 18:00 UTC on 1 October 2017 (Fig. 1c), i.e., 40 minutes to 2 hours before the observation, hint at a fast dispersion due to the different initialization time along the cross section and especially due to the strong wind shear in the jet stream. Going backward in time to 18:00 UTC on 30 September 2017 (approx. 24 h before the observations), the trajectories initialized near the jet core (predominantly in MIX-1 and MIX-2) were located at the North American east coast (Fig. 4a). On the contrary, the transport in the northern part of MIX-2 and in all TRO classes proceeds more slowly. Some of

the southerly air parcels are situated close to the former TC Maria (NOAA/NHC, 2019), which moved as a hurricane from the Caribbean towards the US east coast from where it turned eastward, weakened and underwent extratropical transition until 18:00 UTC on 30 September (see Maria's best track and locations in Fig. 4). Another 24 h earlier, at 18:00 UTC on 29 September (Fig. 4b), the air parcels are deflected north and south of a blocking anticyclone over the western US. A smaller part of MIX-1 and most of MIX-2 move with STRA and MIX-3 in the northward deflected polar jet stream, while the other

part of MIX-1 and MIX-2 is transported south of the block together with the TRO classes (see also Fig. 2). At 18:00 UTC on 28 September (Fig. 4c), TRO-1, TRO-2 and TRO-3a can be assigned to transport from the subtropical eastern North Pacific and western North Atlantic, where they are located in the surrounding of tropical storms Maria and Lee. TC Lee (NOAA/NHC, 2018) re-intensified to hurricane strength the days before and was located east of TC Maria. About five days before the observation at 18:00 UTC on 26 September (Fig. 4d), MIX-1 is embedded in a meandering polar jet stream over the North

Pacific before it splits downstream over the US. MIX-2 is partly embedded in the jet stream but also found further north in a blocking anticyclone in the central North Pacific. Figures 4e and f show that 7 to 10 days before the observations, STRA and a small part of MIX-2 are in the high Arctic within a cyclonic vortex located slightly off the pole while the remaining and larger part of MIX-2 is transported from Russia towards the Pacific anticyclone (see also Fig. 2). At these early times, MIX-1 experiences zonal advection in the jet stream from regions as far as western Africa. At the same time, tropospheric air observed

on the southern side of the jet stream is dispersed in the subtropics and tropics. TRO-1 and TRO-2 dominate over the eastern Pacific and TRO-3a over the western Atlantic (see also Fig. 2). TRO-3b, i.e., the tropospheric air north of the jet stream, shows the slowest transport of the TRO classes with origins over the western North Pacific.

Based on these transport characteristics, seven areas are defined that capture the major weather systems over the 10-day period (Fig. 4a and Table 1) to further investigate their influence on the observed $H_2O$ and $O_3$ distribution. The Atlantic polar jet

(APJ) region covers the southwesterly polar jet stream over the North Atlantic, which is the dominant feature in the approx. two days before the flight (see Figs. 1 and 4). Upstream, the US anticyclone (USAC) region captures the blocking anticyclone that slowly moves from the western North Pacific to the eastern US. Adjacent to USAC, the high Arctic (HAR), the Pacific anticyclone (PAC), and the Pacific polar jet (PPJ) regions cover the dominant weather systems further upstream. Air in the tropics is attributed to the Pacific tropical (PTR) and the Atlantic tropical (ATR) regions. The ATR's boundary is selected to

cover the entire track of both tropical storms.





**Table 1:** Regions covering the dominant weather systems that were observed upstream of the lidar observation in a 10-day period.

| Region | Abbreviation | Latitude range | Longitude range |
|---|---|---|---|
| High Arctic | HAR | 75°−90° N | 360°−0° W |
| Pacific anticyclone | PAC | 55°−75° N | 250°−150° W |
| Pacific polar jet | PPJ | 30°−55° N | 320°−130° W |
| US anticyclone | USAC | 35°−80° N | 160°−75° W |
| Atlantic tropics | ATR | 0°−50° N | 90°−20° W |
| Pacific tropics | PTR | 0°−30° N | 250°−90° W |
| Atlantic polar jet | APJ | 40°−60° N | 80°−10° W |



**Figure 4:** Positions of trajectories (starting at 18:00 UTC on 1 October 2017) at 18:00 UTC on (a) 30 September (-24 h), (b) 29 September (-48 h), (c) 28 September (-72 h), (d) 26 September (-120 h), (e) 24 September (-168 h) and (f) 21 September, (-240 h) coloured by air mass classes (see Fig. 1). All panels show geopotential height at 250 hPa (black contours, ΔZ = 160 gpm), mean sea level pressure (pink contours, Δp = 10 hPa for p < 1005 hPa) and best tracks (thick black line) and positions (white dot) of tropical cyclones Maria (18:00 UTC on 16 September to 18:00 UTC on 30 September) and Lee (18:00 UTC on 14 September to 00:00 UTC on 30 September). Panel (a) shows regions used for a weather system classification (see text for details and Table 1).




Figure 5 depicts the residence time within the individual weather system regions, again projected forward to the lidar cross section and the T–T space. Air masses that are residing in HAR for about six days correspond to increased O$_3$ VMRs and extend as two small-scale filaments into the tropopause fold (Fig. 5a). Highest O$_3$ VMR (> 420 ppb), i.e., pure stratospheric air, remained in HAR the longest while MIX-2 has a slightly shorter residence time in HAR (Fig. 5b). The filamentary structure
in the tropopause fold is bound by air that has an increased residence time (2–5 days) in the Pacific anticyclone (PAC, Fig. 5c). Obviously, the polar air masses at lower levels lose some of their stratospheric characteristic during descent in the tropospheric fold where they are surrounded by air from the midlatitude PAC. An increased time in PAC is clearly associated with MIX-2 connecting high VMRs of tropospheric H$_2$O and stratospheric O$_3$ in T–T space (Fig. 5d). The stratospheric air close to the jet stream and between the filaments extending into the tropopause fold is not affected by PAC and HAR, but
instead transported for 2–7 days within the wavy Pacific jet stream (PPJ, Fig. 5e). These air masses form the MIX-1 and MIX-2 mixing lines at relatively low H$_2$O and O$_3$ (Fig. 5f). Additionally, TRO-3b air has initially travelled in the PPJ (see Fig. 4e and f). Further downstream, stratospheric and mixed air remain for 2–4 days in USAC, obviously independent of whether the transport occurs north or south of the block. Only one filament with increased time in USAC is apparent close to the jet axis, which corresponds to air that is trapped in the block (see Fig. 4e, f). Slowly moving TRO-3b air was located for
the longest time in USAC. Interestingly, a small part of MIX-1 at highest altitudes is not influenced by HAR, PAC, PPJ or USAC and therefore must be related to transport from the tropics.

Tropospheric air was not affected by HAR, PAC or PPJ. All descending air that that was not previously lifted from the PBL (see Fig. 3g and i) and a small area with lifting close to the jet stream (at 10 km altitude and a distance of 600 km) remained only briefly in USAC when the air was transported over the southern US and Mexico (see Fig. 4b, c). The lifted air merges
with the upper-tropospheric descending air downstream of USAC at a late time when the parcel locations are more coherent (see Fig. 4c). Figure 5i indicates that most of the lifted air (Fig. 3g) resided in ATR for almost the entire 10-day period. The southern part of the tropospheric descending air was located 2–3 days in ATR (note that ATR extends into the midlatitudes) and also 2–3 days in PTR (Fig. 5i, k). In contrast, the descending air closer to the jet stream was only for 2 days in ATR and for an extended period of 7–9 days in PTR. Figure 5j and l highlight that TRO-1 and TRO-2 air with a longer residence time
in PTR has lower O$_3$ VMR compared to the air that longer resided in ATR.



**Figure 5:** Geometrical (left column) and T–T space (right column) distributions (as in Fig. 1) coloured by the time the 10-day backward trajectories were located in weather system regions (see Table 1): (a, b) HAR, (c, d) PAC, (e, f) PPJ, (g, h) USAC, (i, j) ATR and (k, l) PTR. Coloured dots mark starting locations of trajectories shown in Fig. 7.



### 3.4 Specific aspects

The previous section described the role of weather systems for the observed $O_3$ and $H_2O$ distributions. Two aspects will be examined now in more detail: (i) the tropospheric lifting processes to clarify the role of TCs and convection; and (ii) the merging of polar $O_3$-rich air with midlatitude air to assess how it affected the stratospheric $O_3$ distributions.

We already indicated that TRO-3a air ascends immediately before the flight, while lifting of TRO-1 and TRO-2 air takes place earlier and further south in the subtropics, predominately over the North Atlantic. Figure 6 shows the location of the strongest ascents (>50 hPa in 6 hours) of the trajectories that were initialized in tropospheric air (PV < 2 PVU). Strong ascents are found close to the centre of former hurricane Maria one day before the observation, at 18:00 UTC on 30 October (Fig. 6a). At that time, Maria was categorized as an extratropical storm in agreement with the frontal clouds in the satellite imagery (Fig. 6b). The lifting can be attributed to a warm conveyor belt-like ascent of moist midlatitude air to altitudes of 9–10 km along the lidar cross section. Three days before the observation, at 18:00 UTC on 28 September (Fig. 6c, d), the ascents are found further south and can be attributed to TC Maria (at hurricane stage), TC Lee (at tropical storm stage) and an area northeast of the Yucatán Peninsula that also shows increased convective activity in the satellite imagery. At 18:00 UTC on 26 September ascents are again related to the TCs (Fig. 6e, both at hurricane stage). Additionally, two regions of ascent over the Pacific, one over the ocean and one near the Mexican coast, coincide with strong convective activity in the satellite imagery (Fig. 6f) related to the northward shifted Intertropical Convergence Zone (ITCZ). At 18:00 UTC on 24 September (Fig. 6g, h), most Atlantic ascents are surrounding hurricane Maria. In addition to the two aforementioned areas in the Pacific, a region of ascent occurs further north at the Mexican Pacific coast that is potentially related to a far southward extending midlatitude frontal zone.[1] Lifting occurs continuously and the time steps in Fig. 6 are selected to illustrate the main ascending mechanisms. In conclusion, air is lifted over the Pacific due to convection in relation to the ITCZ 5–10 days before the flight while the Atlantic is dominated by two TCs that are lifting air during the entire 10-day period.

Figure 7 shows eight trajectories during the 5 days prior to the observations that are initialized at 360 K along the lidar cross section (see dots in Figs. 1, 3 or 5). Due to the tilting of the isentrope in the upper-level frontal zone, pressure at the starting points increases towards the south (Fig. 7a). All trajectories descend in the approx. 36 h before the observation. The white and grey trajectories start in the troposphere ($O_3$ VMR < 100 ppb). While the white trajectory descends from the PTR (see also Figs. 3 and 5), the grey trajectory experienced transport from ATR. The slightly increased $H_2O$ (VMR = 10 ppm) of the grey trajectory at the observation location compared to the surrounding (Fig. 1a) is related to lifting that occurred before the considered 120-h time window. However, the trajectory is still encircling tropical storm Maria on days 4–5 (Fig. 7b). The impact of lifting on the upper-tropospheric $H_2O$ distributions is highlighted by the heterogeneous distribution of specific humidity at 200 hPa (Fig. 7b–d) over the Atlantic with increased values to the north of the TCs. Trajectories in MIX-1 (red and orange) show the transport north and south of the USAC and suggest pathways in the PPJ region earlier on (see Fig. 5e

---

[1] Please note that further north over the US, tropospheric air is lifted at same frontal zone ahead of the upper-level cut-off cyclone and can be attributed to TRO-3b air that was observed north of the jet stream (see discussion in Sec. 3.2).





and f). Although located at slightly different altitudes, these four air parcels come relatively close together on the leading edge of the trough at -24 h (Fig. 7d) in a region of strong isobaric $H_2O$ and wind gradients.

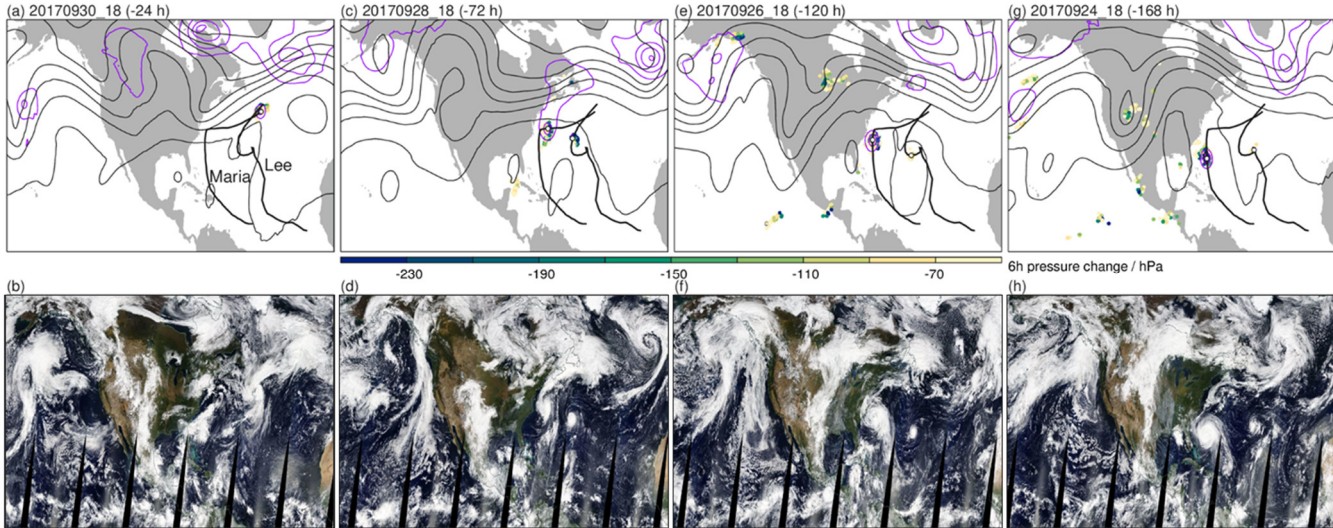

**Figure 6:** (a–d) 6-h pressure change (coloured dots, >50 hPa/6 h) of tropospheric (> 2 PVU) air masses superimposed by geopotential height at 250 hPa (black contours, $\Delta Z$ = 160 gpm), mean sea level pressure (pink contours, $\Delta p$ = 10 hPa for p < 1005 hPa) and best tracks (thick black line) and positions (white dot) of tropical cyclones Maria and Lee. (e–h) Terra MODIS Corrected Reflectance satellite composite images (taken from NASA Worldview through https://worldview.earthdata.nasa.gov).

Figure 7 also informs about the formation of $O_3$ filaments extending into the observed tropopause fold. The yellow and dark blue coloured trajectories start within the filament of increased $O_3$ VMR (350 and 425 ppm, see Fig. 1b), while the green and light blue ones start in between and north of it, i.e., in regions of lower $O_3$ VMR (320 ppm). At 06:00 UTC on 27 September (-108 h, Fig. 7b) the four trajectories are in polar regions. The $O_3$-rich air (yellow and dark blue) is located further north in a region of lowest $H_2O$ concentrations, i.e., in the HAR, which was the dominating weather system also before the considered 5-day time period (see Fig. 5d). With the northward propagation of USAC, the Arctic dry air gets squeezed over northern Canada at 18:00 UTC on 28 September (-72 h, Fig. 7c) when stratospheric trajectories are located on the anticyclonic shear side of the jet stream. Downstream, the southeastward propagating trough gets narrower and, in a zone of strong shear and deformation, the dry air gets stretched in southwest to northeast direction until 18:00 UTC on 30 September (-24 h, Fig. 7d). At this time the stratospheric trajectories lie within the trough and show a strong descent afterwards. Interestingly, differential advection within the trough causes the green trajectory to arrive within the $O_3$-rich air (blue and yellow) causing the observed filamentation.

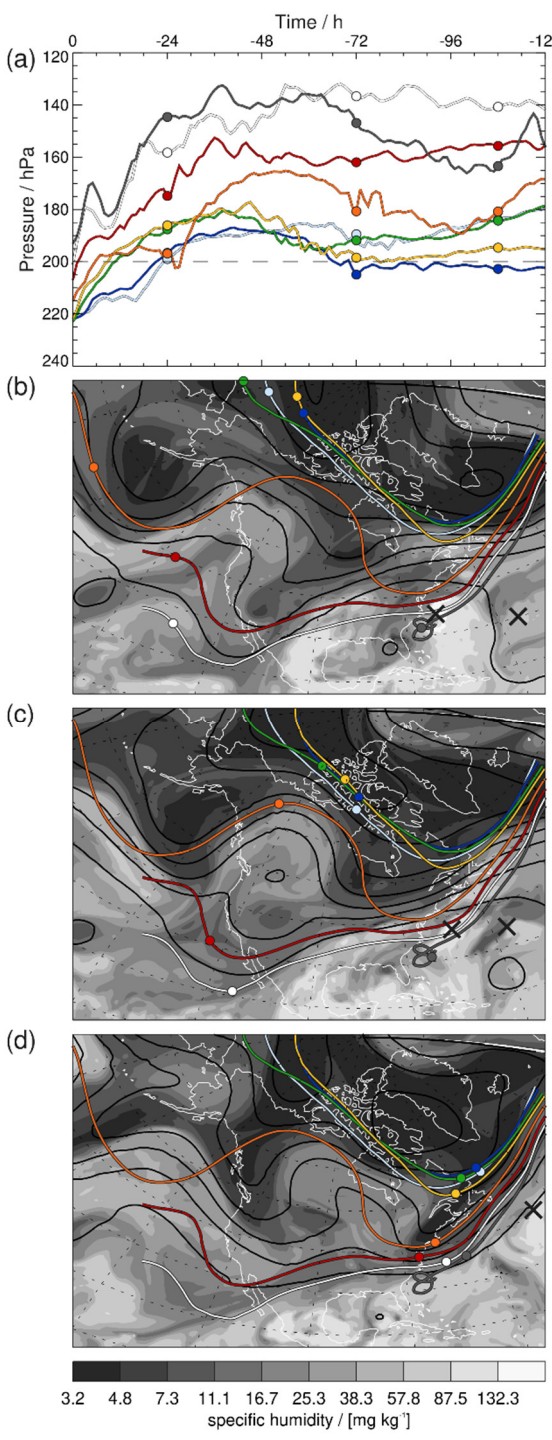

370

**Figure 7:** (a) Five-day pressure evolution along selected trajectories starting at 360 K (see coloured dots in Figs. 1, 3 and 5). (b–d) Trajectory pathways for 0 to -120 h (coloured lines) and individual locations at (b) 6:00 UTC on 27 September 2017 (-108 h), (c) 18:00 UTC on 28 September (-72 h) and (d) 18:00 UTC on 30 September (-24 h) together with 200 hPa geopotential height (black contours, ΔZ = 160 gpm) and specific humidity (grey shading, mg kg$^{-1}$). Black crosses mark locations of tropical cyclones Maria and Lee.



### 3.5 Mixing processes

This section investigates the impact of turbulent mixing processes associated with the tropospheric weather systems and their effect on $H_2O$ and $O_3$ distributions in the ExTL. Figure 8a and b show distributions of Ri and TI at 18:00 UTC on 1 October, i.e., at about the time of the measurements. Both criteria of Ri < 2 and TI > 8 x $10^{-7}$ $s^{-1}$ capture regions of increased vertical wind shear. These regions are located in the fold below the jet maximum and above, in two areas that are separated by reduced shear in-between. For the lower of the two, directly above the jet maximum, Ri points to turbulence on the tropospheric and, TI to turbulence on the stratospheric side. Enhanced turbulence occurs in the upper-level frontal zone and in the lower tropospheric front (discernible by the tilted isentropes), while turbulence is reduced at the level of maximum winds. This agrees well with the conceptual model in Shapiro (1976).

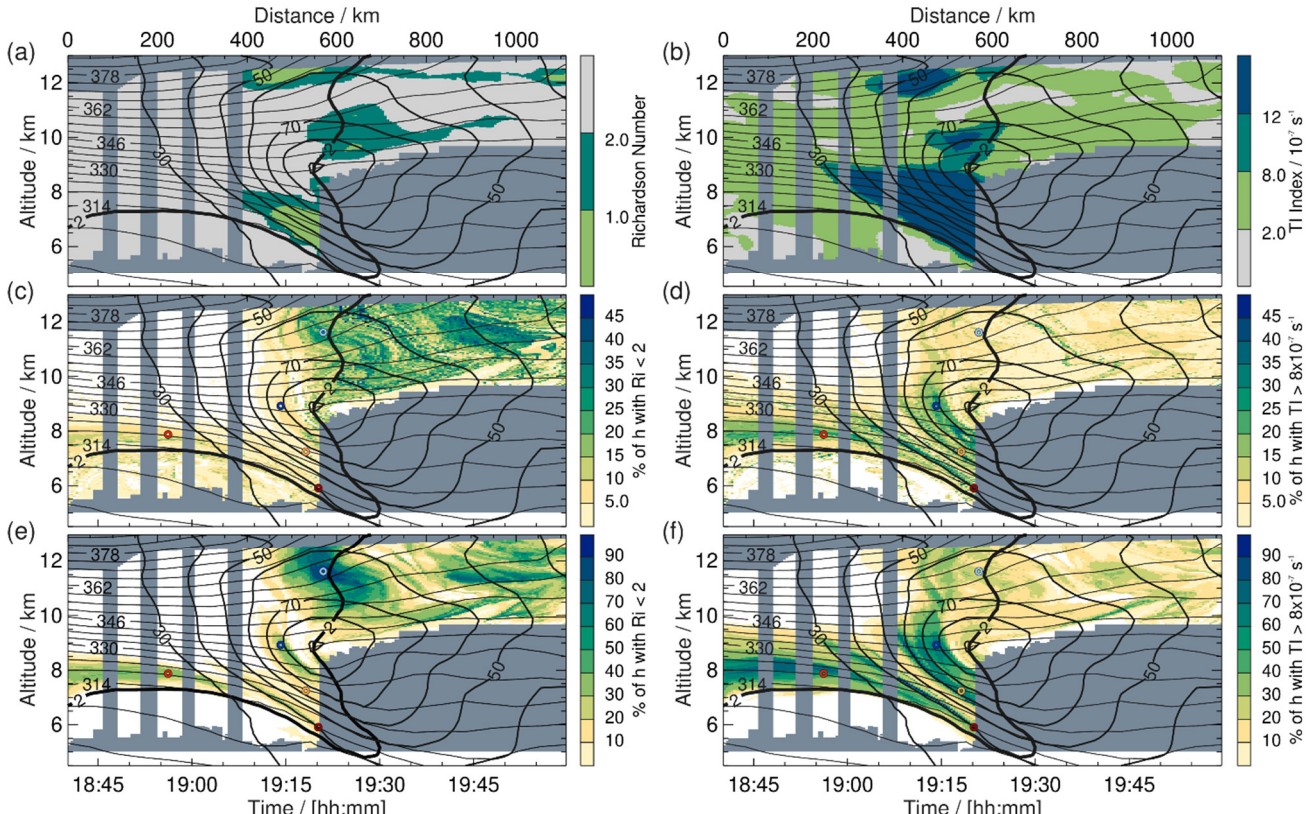

**Figure 8:** Instantaneous distribution of (a) Richardson Number (Ri, colour shading) and (b) Ellrod-Knapp turbulence index (TI, colour shading) at 18:00 UTC on 1 October 2017. Percentage of time steps with (c) Ri < 2 and (d) TI > 8 for the entire 10-day period and (e) Ri < 2 and (f) TI > 8 x $10^{-7}$ $s^{-1}$ for 48 h before the observations (18:00 UTC on 1 October to 18:00 UTC on 29 September 2017). Coloured circles in (c–f) mark starting positions of trajectories as shown in Fig. 10.

Figure 8c and d display the percentage of the 240 hourly timesteps along each 10-day backward trajectory that experience turbulence according to Ri < 2 and TI > 8 x $10^{-7}$ $s^{-1}$. Although both parameters hint at a similar area along the cross sections, the magnitudes largely differ. Ri shows a high number of time steps (up to 40 %) in the upper-troposphere (TRO-1 and





TRO-2) and in MIX-1, but also in MIX-2 (up to 10 %). In contrast, TI highlights mixing events in MIX-2 and much less in the upper troposphere. STRA exhibits no mixing events during the 10-day transport in both cases. Similar patterns emerge when only the 48 h before the observations are considered (Fig. 8e, f) and thus many turbulence events occur in this period.

This is further investigated by projecting the fraction of turbulent time steps in 48 h intervals to T–T space (Fig. 9). It clearly shows an enhanced turbulence in the mixed air in the 48 h before the observations for both Ri and TI. Ri highlights turbulence (up to 70 %) in MIX-1, TRO-1 and TRO-2 as well as to a smaller degree mixing in MIX-2. In contrast, TI shows high values in all MIX classes with an emphasis on MIX-2 and MIX-3. Interestingly, the region with the highest number of turbulence events based on the TI (Fig. 8f) approximately follows the air with the highest distance from the undisturbed tropospheric and

stratospheric air in the T–T space at the centre of MIX-2 [see Fig. 9a, f and Fig. 7 in Schäfler et al. (2021)], which is considered as air that experienced the strongest mixing of tropospheric and stratospheric air (Kunz et al., 2009).

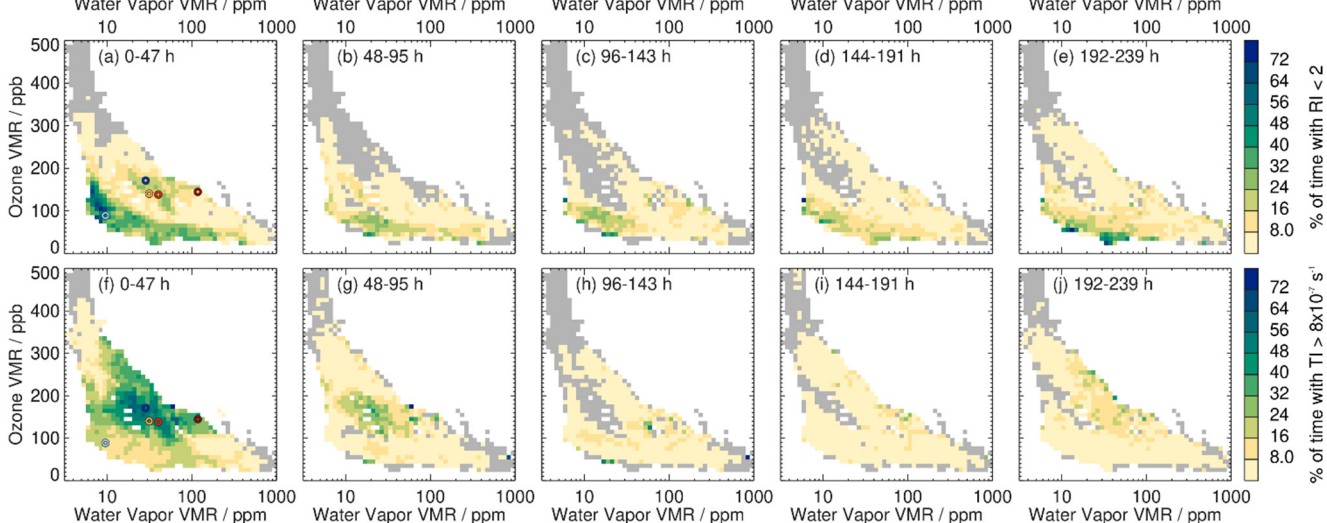

**Figure 9:** Bin average T–T distribution of the percentage of time steps with (a–e) Ri < 2 and (f–j) TI > 8 x 10⁻⁷ s⁻¹ (colour shading) for 48 h
time intervals: (a, f) 0–47 h, (b, g) 48–95 h, (c, h) 96–143 h, (d, i) 144–191 h and (e, j) 192–239 h. Coloured circles in (a, f) mark starting positions of trajectories as shown in Fig. 10.

In order to better understand where turbulence occurs and how it affects different regions of the ExTL, Fig. 10 shows vertical profiles of Ri and TI along six selected 4-day backward trajectories that are all characterized by increased turbulence (see dots in Figs. 8 and 9). The light blue trajectory (Fig. 10a, b) starts in MIX-1 (see Figs. 8 and 9). The dark blue, orange and light red

trajectories (Fig. 10c–h) are initialized in different parts of MIX-2 and the dark red (Fig. 10i, j) in MIX-3 (see Figs. 8 and 9). To inform about static stability and wind shear, Fig. 10 additionally shows the evolution of potential temperature and wind speed.



**Figure 10:** Vertical profiles along 96 h backward trajectories of Ri (left column, colour shading) and TI (right column, colour shading) superimposed by horizontal wind speed (beige contours, in m s$^{-1}$), potential temperature (grey contours, in K) and dynamical tropopause (2 PVU, thick black contour). The vertical position of the trajectories is colour-coded by the starting position as shown in Figs. 8 and 9. Coloured dots on top of each panel mark locations with Ri < 2 or TI > 8 x 10$^{-7}$ s$^{-1}$.

The light blue air parcel (Fig. 10a, b) that is initialized at high levels close to the tropopause (see Fig. 8) experiences no turbulence until it overpasses the Sierra Madre mountains in Mexico at -48 h. Thereafter, it is transported in the southwesterly flow at high wind speeds in the upper part of the jet stream close to the tropopause. Both turbulence indicators similarly show increased turbulence potential above the wind maximum in regions of strong vertical wind shear. However, the TI values are





often smaller than the applied threshold leading to a reduced number of diagnosed turbulent time steps. The proximity to the tropopause with likely increased mixing activity fits the uniformly mixed chemical characteristic across the tropopause in the

upper part of the jet stream, as already described in Schäfler et al. (2021).

The dark blue air parcel (Fig. 10c, d) is initialized slightly north of the wind maximum and transported in the northern part of USAC in the anticyclonic flow (see also Fig. 7) and further upstream in the PPJ. For almost the entire time the air parcel experienced strong winds above the tropopause. At -69 h the trajectory passes over the Canadian Rocky Mountains. Until -24 h it was located in the upper part of increased winds in the anticyclonic flow. The TI is increased, which might be related

to the location just north of the jet stream where Jaeger and Sprenger (2007) found the highest climatological values of TI. At -24 h, when the air parcel is reaching the trough at the US east coast (see Fig. 7d), the tropopause below suddenly drops in altitude and the trajectory is subsequently located at the level of maximum winds in the APJ with little turbulence in both metrics (Fig. 10c and d). The increased mixing activity before -24 h agrees with the well-mixed ExTL diagnosed in this region in Schäfler et al. (2021). In turn, the region further south on the cross section that shows low turbulence activity (Fig. 8c–f)

agrees with the discontinuity across the tropopause at these altitudes (Schäfler et al., 2021).

The orange air parcel (Fig. 10e and f) that starts below the maximum winds at the centre of the downward sloping tropopause (see Fig. 8) exhibits a similar transport as the dark blue one (not shown) but is located slightly north until -60 h (not shown), which is reflected in reduced wind speeds. It crosses some regions of enhanced turbulence probability in the anticyclonic flow and especially after it is entering the trough at the US east coast at -30 h. This is accompanied by a decrease in tropopause

altitude and a transport below the jet maximum afterwards. The fact that this region is more pronounced in TI than in Ri points to enhanced deformation which is plausible in the trough region (see Sec. 3.4). Just before the observations, Ri also predicts turbulence in a region of strong vertical shear. The mixing of stratospheric air with surrounding tropospheric air from below or the sides may have caused the diagnosed mixed air in this region (Schäfler et al., 2021).

The light red trajectory (Fig. 10g and h) starts in MIX-1 but further away from the jet stream. This air parcel was also

transported in the anticyclonic flow in the northern part of USAC (not shown), however, it already reaches the trough at -48 h when it is located further west than the so far discussed trajectories (Fig. 7c and d). Further downstream between -42 and -24 h, it is situated below the Atlantic jet and experiences turbulence, which is again more emphasized in TI. The last 18 h before the observations exhibit little wind and turbulence in accordance with the starting position. Thus Fig. 10g and h indicate that the formation of the ExTL further north was influenced by very similar processes near the tropopause below the jet stream.

Finally, the dark red air parcel (Fig. 10i and j) starts in MIX-3, which was described as a region of recent mixing between TRO-3b and MIX-2 (Schäfler et al., 2021). Transport proceeds much slower and the air parcel entered the Atlantic jet stream comparably late and downstream of the coast of Newfoundland (Fig. 4a, b). Ri and TI agree on the increased turbulence activity in the late phase below the APJ, however, they disagree further upstream where TI shows additional turbulence. The recent mixing is well conceivable as the trajectory was very close to the tropopause in this late phase. A reason why this process

appears as a separated mixing regime in T–T space might be related to the increased $H_2O$ VMR in TRO-3b, which was only involved in mixing in the lower part of the tropopause fold.



## 4. Discussion

The ExTL, which is a part of the UTLS, has been intensively studied in the past decades (Gettelman et al., 2011) as it is of key relevance for the Earth's weather and climate. Tropospheric and stratospheric dynamical processes acting on different time-scales interact with chemistry and thus determine the characteristics of this mixed layer and its strong seasonal and short-term variations. Many past studies have focused on exchange processes across the tropopause, origin and pathways of observed air masses and the characterization of the shape of the ExTL. In the presented case study, we demonstrate how strongly the fine-scale trace gas distribution in the UTLS/ExTL is related to interacting tropospheric weather systems, which shape transport and mixing on synoptic time scales. The presented lidar observations from one flight during the WISE field campaign uniquely reveal the distribution of $O_3$ and $H_2O$ in the UTLS across a North Atlantic jet stream.

The midlatitude and tropical origin of the tropospheric air agrees with complementary work on other trace gases along the same flight track but at lower flight altitudes. Lauther et al. (2021) linked in situ observed, exceptionally low concentrations of chlorine containing very short-lived substances to fast vertical transport in TC Maria from weak emission source regions in the tropical Atlantic PBL. Rotermund et al. (2021) found increased concentrations of bromine, which they connected to lifting by TCs and convection. During the major hurricane season (August–October) such TCs, which often undergo extratropical transition, are not rare and may be one reason why climatological distributions of $H_2O$ and $O_3$ also suggest mixing with tropical dry air (Hegglin et al., 2009). Vogel et al. (2011) also observed a separation of mixing lines in $CO$–$O_3$ in situ observations for a midlatitude jet stream crossing and found contributions of low-latitude air ($< 30°$ N) in tropospheric air equatorward of the jet. In contrast to our case, the mixed air of tropical character was a result of a tropospheric intrusions, i.e., a cut-off of tropospheric air which moved into the midlatitudes and subsequently got mixed with stratospheric air (Homeyer et al., 2011). Other WISE observations did not show tropical air masses. Generally, individual flights reveal a large variability in terms of $H_2O$ and $O_3$ distributions but also the related transport pathways. However, as the presented $H_2O$ and $O_3$ observations are well within climatological distributions and the distribution in the ExTL resembles the schematic view of the UTLS (Gettelman et al., 2011), we consider this case to be fairly representative for the autumn season [see also discussion in Schäfler et al. (2021)].

The presented approach to trace different meteorological parameters and derived turbulence diagnostics along trajectories allows us to analyse transport and mixing characteristics for all observation locations and to project the information into T–T space. This combined approach is complementary to studies applying more sophisticated chemistry transport models that parameterize mixing based on shear and deformation of the large-scale wind (Konopka and Pan, 2012; Konopka 2007; Vogel et al. 2011). Although the presented turbulence indicators can only determine a potential for mixing and they cannot tell anything about the changes of the chemical composition due to turbulence, the high frequency of turbulence in the mixed air in T–T space points to a decisive role of mixing in the jet stream. The importance of these short-term processes for the ExTL agrees with Konopka and Pan (2012), who found increased numbers of mixing events along three-day trajectories close to the jet stream and in the adjacent ExTL at low tropopause altitudes. They also showed that an artificially removed ExTL





regenerates partly within three days. If chemical transport models were capable of resolving the DIAL $H_2O$ and $O_3$ distributions, which seems reasonable because of the importance of synoptic scale transport processes, an analysis of the air mass transformations would be valuable to study the relevant time scales that influence the ExTL in greater detail.

The presented results further point to the complexity of mixing lines in T–T space that are often interpreted as physical mixing between the tropospheric and stratospheric background concentrations. Principally, this is correct but not in the sense that the

observations obtained along flight trajectories or lidar cross sections are mixed at the time they are observed. As a key result of this study, we illustrate the non-local and transient character of mixing, which makes the interpretation of mixing lines more challenging and documents the advantage of the present approach. Differential advection can quickly change the involved, neighbouring air masses and impact the mixing state as for example demonstrated for the mixed air class MIX-3. Overall the identified regions of turbulent mixing agree well with regions that indicated homogeneous transitions in T–T space in Schäfler

et al. (2021), which indicates a certain persistence of neighbouring air masses, at least during the two days of increased mixing in the North Atlantic jet stream. The vertical profiles of the turbulence indicators give additional insight in the location relative to the jet stream and the processes driving CAT. Although we found significant differences between the two used turbulence metrics, they often highlight comparable regions. A more thorough analysis of the sensitivity to the selected thresholds is beyond the scope of this study. However, a consideration of other metrics may help to better understand the relation of

turbulence and mixing (Dörnbrack et al., 2022)

The collocated WISE $H_2O$ and $O_3$ DIAL observations were also used to better investigate the vertical structure and origin of the lower-stratospheric moist bias in ERA5 reanalyses (Krüger et al., 2022). An increased bias in the ExTL indicated an overestimation of $H_2O$ transport into the lower stratosphere. The results here suggest that the NWP bias is formed on time scales of a few days and connected to mixing in the polar jet stream. Additional collocated $H_2O$ and $O_3$ observations in other

meteorological situations could help to distinguish the role of different mixing processes for the formation of the bias, e.g., convective overshooting moistening the lower stratosphere (Homeyer et al., 2014).

The study assumes that the trajectory calculations are reasonably accurate to investigate the transport over a period of 10 days. Trajectories were initialized exactly from the locations of the observation, however, due to computational costs and data storage issues they were started in 5 minutes intervals. The one-hourly analyses provided by ERA5 are certainly an improvement to

previously often applied combined analysis and forecast data, however, ERA5's spatial resolution (31 km) is comparatively low, which might be crucial for resolving small-scale weather systems, e.g., convection. Therefore, it is astonishing how well even small-scale variability can be explained by differing transport pathways. Ascent in TCs or tropical convection that is known to provide increased uncertainty from the driving wind fields is represented reasonably well as could be verified with satellite imagery even 8–10 days before the flight.



## 5. Summary and conclusions

Based on a recent characterization of mixing in the UTLS using two-dimensional cross section of $H_2O$ and $O_3$ by an airborne nadir pointing DIAL (Schäfler et al., 2021), this study investigates the role of tropospheric weather systems for the observed trace gas distributions. In a novel approach, lidar observations across a midlatitude jet stream obtained during the WISE campaign on 1 October 2017 are combined with analyses in tracer–tracer (T–T) space and 10-day backward trajectories from all observation locations. Beside meteorological parameters characterizing the transport, indicators for turbulent mixing are traced along the individual trajectories and projected into T–T space. We find that the formation of $H_2O$ and $O_3$ filaments in the troposphere and stratosphere, the high variability of tropospheric $H_2O$ and the formation of the ExTL mixing layer can, to a large extent, be explained by transport and mixing associated with tropospheric weather systems on synoptic time scales.

Considering the synoptic evolution, seven characteristic regions containing the dominant weather systems are identified. Before the time of the observations, the air is transported in the Atlantic polar jet stream (APJ) and exhibits a split flow around an anticyclone over the US (USAC). Further upstream the air parcels are partly involved in Pacific (PTR) and Atlantic tropical weather systems (ATR), i.e., tropical cyclones (TCs) and ITCZ convection. Additionally, the Pacific polar jet stream (PPJ), an anticyclone over the midlatitude Pacific (PAC) and flow from the high Arctic (HAR) played an important role. The main influence of the diagnosed weather systems for the highly variable $H_2O$ and $O_3$ distribution in stratospheric, tropospheric and mixed air is summarized as follows:

- *Lower stratosphere (PV > 2 PVU):* Increased $O_3$ values that stretch as a filament across the stratospheric background and the mixed ExTL air into the tropopause fold beneath the jet stream are associated with air from HAR. The longest residence times in HAR are correlated with highest $O_3$ VMRs. The lower the altitude of the filament, the less of its stratospheric character is preserved. The air surrounding the filament possesses relatively high VMRs of $H_2O$ and $O_3$ and originates from the PAC. Both air masses merge in the anticyclonically curved and northward deflected jet stream associated with the USAC. The filamentation observed in the lidar cross section is caused by differential advection in a deepening and narrowing trough over eastern Canada.

- *Upper troposphere (PV < 2 PVU):* The tropospheric air south of the jet stream is strongly influenced by weather systems in ATR and PTR, i.e., tropical convection related to the ITCZ as well as the TC's Lee and Maria. Although tropospheric air masses share a transport from southwest, the moist air (VMR $H_2O$ > 100 ppm) and the dry air above (VMR $H_2O$ < 100 ppm) have different origins in the midlatitudes and tropics, respectively. The dry air is mainly characterized by a descent from tropical origins in the ATR or PTR. However, strong upward transport in the TCs continuously injects $H_2O$ into the upper troposphere and forms several filaments of increased $H_2O$, which are embedded in the descending air. The surrounding lower $H_2O$ VMRs are likely caused by lifting to colder temperatures in the ITCZ, however, this must have happened before the considered 10-day period. The increased turbulence activity in the tropical air, which is likely related to low static stability in the upper troposphere, may have reduced some of the $H_2O$ contrasts. The moist air that forms a flat and southward extending layer originates from lifting of midlatitude air immediately before the observations, again in connection with TC





Maria that already underwent extratropical transition. Tropospheric air to the north of the jet stream joined the air in the APJ rather late after being transported in the northern part of the USAC.

• *Mixed air:* In the two days preceding the flight the mixed air is located mainly in the APJ and the associated wind shear causes a fast dispersion of air parcels. A high correlation of mixed air in T–T space is found with two indicators for CAT, the Richardson number (Ri < 2) and the Ellrod-Knapp turbulence index (TI > 8 x $10^{-7}$ s$^{-1}$), which points to an increased importance of turbulent mixing during this time period. While Ri emphasizes turbulence in the upper part, TI highlights turbulence within the entire ExTL. The highest number of turbulence events is found for air with the highest distance to the

undisturbed tropospheric and stratospheric background in the T–T phase space which further suggests a strong link to the diagnosed turbulence. Mixed air in the upper part of the jet stream at lower $H_2O$ and $O_3$ VMRs experiences turbulence in regions of strong vertical wind shear in the upper part of the APJ and further upstream over the US and Mexico south of the USAC. Mixed air in the lower part and north of the jet stream is impacted by turbulence in the anticyclonically curved jet stream related to the USAC and, subsequently, in the wind-shear zone below the APJ. It is additionally demonstrated that,

below the APJ and immediately before the observations are taken, midlatitude tropospheric air north of the jet stream mixes with ExTL air above.

Although the observations from this case study resemble common tracer distributions for the autumn season, the presented case study can only provide a snapshot of the UTLS. However, it is remarkable how well the occurrence of tropospheric weather systems over a period of 10-days and their impact on transport pathways and on mixing processes can explain the

observed high $H_2O$ and $O_3$ variability. Thus, the large case-to-case variability in UTLS trace gas distributions found in remote-sensing and in situ observations and also the large variability of transport pathways can be attributed to a correspondingly large variability of tropospheric weather systems and their complex interaction. The presented analysis of lidar observations in physical space and in tracer space combined with Lagrangian diagnostics appears to be a useful approach to explain the small-scale trace gas variability in the context of transport and mixing governed by tropospheric weather systems.

**Data availability.** The lidar data used in this study is available through the HALO database (https://halo-db.pa.op.dlr.de/). We are grateful to ECMWF granting access to the full-resolution ERA5 data. The ETH access to the ECMWF data is provided by the Swiss National Weather Surface (MeteoSwiss).

**Author contribution.** AS, MS and HW designed the study. AS performed the analysis, produced the figures and wrote the paper. MS created the setup of LAGRANTO. MW performed the analysis of the DIAL water vapor and ozone data. AF
performed DIAL observations during WISE. MS and HW advised on the analysis, contributed with ideas, helped with the interpretation of the data and commented on the paper.

**Competing interests.** At least one of the (co-)authors is a member of the editorial board of Atmospheric Chemistry and Physics.The authors declare that they have no other conflicts of interest.



**Acknowledgment.** The authors thank the whole WISE team, especially the coordinators Peter Hoor and Martin Riese, for the
great collaboration. We are grateful to DLR for supporting this work in the framework of the DLR project "Klimarelevanz von
atmosphärischen Spurengasen, Aerosolen und Wolken" (KliSAW). Additionally, we thank the German Science Foundation
(DFG) for supporting the HALO contribution to the WISE campaign within the priority program SPP 1294 HALO. AS
acknowledges the support of the transregional collaborative research center (SFB/TRR 165) "Waves to Weather"
(http://www.wavestoweather.de) funded by the German Research Foundation (DFG). AS thanks DLR for providing the
possibility for a sabbatical stay in the Atmospheric Dynamics group at the Institute of Atmospheric and Climate Science of
ETH Zurich. We thank Andreas Dörnbrack (DLR) for helpful discussions and for his careful review of the manuscript.

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
