# Peer review of "Case study on the influence of synoptic-scale processes on the paired H2O-O3 distribution in the UTLS across a North Atlantic jet stream"

_Atmospheric Chemistry and Physics, 2022_

## Referee Comment (RC2)

**Review of acp-2022-692: Case study on the influence of synoptic-scale processes on the paired H2O– O3 distribution in the UTLS across a North Atlantic jet stream by A. Schäfler et al.**

**Summary**

In the present manuscript the impact of synoptic scale weather systems and turbulent mixing are studied to better explain the distributions of water vapor and ozone in the tropopause region in the vicinity of the jetstream over the North Atlantic. This is done by combining LIDAR observations of these species, taken onboard the HALO aircraft during one research flight of the WISE campaign in 2017, with synoptic weather analysis, tracer-tracer correlations and back trajectories. This study can be regarded as extension of a previous paper of some of the authors where air masses have been classified along the observations. The focus of this paper is more to explain the origin of these air masses along with the atmospheric processes which determine the chemical composition. The authors identify seven major weather systems which affect the pathways of the probed air masses with high latitude air masses causing O3 filaments in the lower stratosphere, tropical cyclones affecting the water vapor distribution in the upper troposphere and the jet stream carrying mixed air masses towards the locations of the measurement.

**Relevance**

This study is a perfect show case to illustrate the complex interplay between tropospheric dynamics and the chemical composition of the UTLS in the extratropics. The study is of high value for the community, first because the DIAL obersvations provide such detailed information which are unique in the tropopause region. Second, the combination with Lagrangian analysis provides further insight in the pathways of air masses in the tropopause region and highlights the non-local but persistent effect of mixing on the chemical composition. Yet alone Figure 4 shows how diverse the air mass origin along a relatively short flight trajectory in the mid-latitudes can be!
I regard the quality of this paper as very high. The line of thoughts is laid out clearly, the question of the paper is expressed in a clear statement, the figures are illustrative and informative, and the conclusions follow from the analysis. I highly recommend publication in ACP. I have only a few minor and technical comments which the authors might address before final publication.

**Suggestions**

*Minor*
1. page 4, line 96 and page 5, line 42: Isn't 10 days a bit long for the synoptic time scale? If I take the synoptic length

scale (1000 – 5000 km) and the characteristic wind speed of about 10-50 m/s, then I end up (optimistically) at around 5-6 days. I am curious whether there is a physical justification for the ten days which I miss here?

2. Table 1 (and associated weather system classification): Is there a specific reason why you refer to the jetstream as polar jet? Is it because of the altitude of the jet, like is it more present at lower levels while the subtropical jet is at higher levels? Has there been an additional subtropical jet (while the polar jet reaches latitudes down to 30°N according to the table).

3. page 5, line 35: Which quantity has been used for the vertical interpolation of the model data onto the aircraft trajectory: pressure, altitude or potential temperature?

4. Page 15, section 3.4, line 341ff: The fate of the O3 rich air at midlatitudes is studied using 8 trajectories. How trustworthy are these individual trajectories? Would it not be better to initialize a cluster of trajectories in the region of the eight starting points (with slightly altered initial positions)?

5. Page 21, line 438: What is meant with "enhanced turbulence probability"?

*Technical*

• Figure 1c: I assume the  geopotential height contours are given in gpm. Which isolines are shown for the surface pressure? Same as in Figure 4?

• Page 6, line 163: "... but also …" => "...and also …"

• Figure 2: The Arctic Circle is difficult to spot (but is in my opinion not too relevant).

• Figure 3g: The figure shows max pressure, and the legend states mean pressure.

• Are the figures vector graphics? When zooming in the figures (since they often contain a lot of information!), they often pixaleted and small letters were diffficult to read.

---

## Author Comment (AC1)

**Response to comments of Referee #1 on**

Schäfler, A., Sprenger, M., Wernli, H., Fix, A., and Wirth, M.: Case study on the influence of synoptic-scale processes on the paired H2O-O3 distribution in the UTLS across a North Atlantic jet stream, Atmos. Chem. Phys. Discuss. [preprint], https://doi.org/10.5194/acp-2022-692, in review, 2022.

We are grateful for the valuable comments, which helped us to improve and complete our manuscript. The response to the individual comments and questions is presented in blue with the corresponding changes to the revised manuscript in *green*.

 (…) The authors provide a very detailed description of the origin of the observed air masses and the influence of synoptic weather systems and mixing effecting it. The paper would probably benefit from a more concise description of the results.

We agree with the reviewer that the paper is detailed in the description of the results, which we found was inevitable to convey a clear picture of the complex transport and mixing processes. We would need to change the focus of the study to be more concise. With regard to the positive feedback from the second reviewer, we decided to keep the level of detail and the complete analysis of the observed cross section. To make the paper easily accessible, we provide the key messages in the introduction, discussion and conclusion sections.

**Minor**

Fig. 1: Why is the air mass TRO-3 separated into two parts for Fig. 1f but not for Fig. 1e (and Fig. 2, although for Fig. 2 it is obvious from the description, which main paths the two parts take, so I think mainly a separation in Fig. 1e would be interesting.)?

Thanks for this suggestion. In the original version of the figure (see Schäfler et al. 2021) we did not separate TRO-3 into two parts. The main motivation for a separate discussion of the transport and mixing behaviour was that the two air masses are spatially separated along the cross section. However, as stated in the caption of Fig. 1 ("*Please note that a separation of TRO-3a and TRO-3b is only possible in the cross section (f) and that both air masses overlap in (e).*"), it is not possible to distinguish them in Fig. 1e as they overlap in T-T space. The clearly different transport for both regions allowed us to summarize TRO-3a and TRO-3b trajectories and to avoid a seventh panel in Fig. 2. We think that in combination with Fig. 4, which distinguishes both regions, this should become clear.

Line 162: Maybe rather: "CAT can modify local gradients of wind (wind shear), temperature (stability) and trace gases (Kunkel et al., 2019)." instead of: "CAT can modify local gradients of winds (wind shear), temperature (stability) but also of trace gases (Kunkel et al., 2019)."

Changed.

Figure 8: It would be good to mention in the caption, that 8c and d do not have the same colour bar as 8 e and f (or use the same, if possible, or maybe the same for Fig. 8 e and f and Fig. 9.)

We added a note to the caption of Fig. 8.: *"Note the variable range for the colour bars for Ri in (c) and (e) as well as for TI in (d) and (f)."*

**Technical**

Line 34: Acronym LS not explained?

As we introduce *"upper troposphere and lower stratosphere (UTLS)"* in line 1, we assume that this is not needed. We leave the discussion to the technical editors and would explain the acronym if needed.

Line 147: Acronym NWP not explained?

Changed.

Line 160: "are an indicator for" or "are indicators for" instead of "are indicator for"?

Changed.

Fig. 2: The "Grey circles mark the Tropic of Cancer and the Arctic Circle." are difficult to see.

We changed the colour of the circles to black and thickened the lines.

Fig. 3: Subscripts of the color bar label are nor readable (zooming in does not help). Higher resolved graphic, vector graphic, and/or larger font size necessary.

This is partly related to the limited quality of the figures in the template version, which we unfortunately haven't noticed at the initial submission. We increased the font size, especially of the subscripts, and we will take care that all figure text are readable in the final typeset version.

Fig. 4 and 6: "pink contours" look rather violet or orchid to me? The white dots marking the hurricane positions are difficult to see.

Changed to *"violet".*

Line 307: Remove double "that" in "All descending air that that was not"

Corrected.

---

## Author Comment (AC2)

**Response to comments of Referee #2 on**

Schäfler, A., Sprenger, M., Wernli, H., Fix, A., and Wirth, M.: Case study on the influence of synoptic-scale processes on the paired H2O-O3 distribution in the UTLS across a North Atlantic jet stream, Atmos. Chem. Phys. Discuss. [preprint], https://doi.org/10.5194/acp-2022-692, in review, 2022.

This study is a perfect show case to illustrate the complex interplay between tropospheric dynamics and the chemical composition of the UTLS in the extratropics. The study is of high value for the community, first because the DIAL observations provide such detailed information which are unique in the tropopause region. Second, the combination with Lagrangian analysis provides further insight in the pathways of air masses in the tropopause region and highlights the non-local but persistent effect of mixing on the chemical composition. Yet alone Figure 4 shows how diverse the air mass origin along a relatively short flight trajectory in the mid-latitudes can be! I regard the quality of this paper as very high. The line of thoughts is laid out clearly, the question of the paper is expressed in a clear statement, the figures are illustrative and informative, and the conclusions follow from the analysis. I highly recommend publication in ACP. I have only a few minor and technical comments which the authors might address before final publication.

We appreciate the careful review of our manuscript and the positive feedback about the relevance of our work. The response to the individual comments and questions is presented in blue with the corresponding changes to the revised manuscript in *green*.

**Minor**

1. page 4, line 96 and page 5, line 42: Isn't 10 days a bit long for the synoptic time scale? If I take the synoptic length scale (1000 – 5000 km) and the characteristic wind speed of about 10-50 m/s, then I end up (optimistically) at around 5-6 days. I am curious whether there is a physical justification for the ten days which I miss here?

We were not precise enough about what we mean with synoptic time scale and why we choose 10 days for the trajectories. We do not consider the 10 days to be a Lagrangian time scale, i.e.

[Figure]

the time needed for an air parcel to pass a typical weather system. Instead, we selected 10 days to be sure to cover the whole lifetime of tropospheric synoptic scale weather systems (including fairly long-lived atmospheric blocks), which is typically illustrated in length and time scale diagrams of weather systems (see left, taken from Stull, 2017)

To clarify this, we first removed the 10 days in connection with the synoptic time scales and changed the sentence in the introduction to *"On shorter synoptic time scales (several days) (…)"*. Second, we added a sentence to Sect. 2.2: *"Note that the 10 days were selected to cover the whole range of lifetimes of synoptic-scale weather systems."*

2. Table 1 (and associated weather system classification): Is there a specific reason why you refer to the jetstream as polar jet? Is it because of the altitude of the jet, like is it more present at lower levels while the subtropical jet is at higher levels? Has there been an additional subtropical jet (while the polar jet reaches latitudes down to 30°N according to the table).

We agree that the naming "Pacific polar jet" neglects the subtropical jet stream over the Pacific at higher altitudes and it is possible that both jet streams merged. We changed the naming to "Pacific jet". The 250 hPa flow (Fig. 4) and the surface pressure, however, show characteristic meridionally disturbed mid-latitude Rossby waves and related surface cyclones, which are the dominant features.

3. page 5, line 35: Which quantity has been used for the vertical interpolation of the model data onto the aircraft trajectory: pressure, altitude or potential temperature?

The interpolation is done in altitude, which is what the lidar profile provides as vertical coordinate. We revised the sentence as follows: *"These reanalysis fields were interpolated bilinearly in the horizontal from the surrounding model grid points, linearly in altitude and linearly in time towards the observation location (Schäfler et al., 2010)."* A more detailed description of the interpolation method is available in the given reference.

4. Page 15, section 3.4, line 341ff: The fate of the O3 rich air at midlatitudes is studied using 8 trajectories. How trustworthy are these individual trajectories? Would it not be better to initialize a cluster of trajectories in the region of the eight starting points (with slightly altered initial positions)?

We choose one particular isentropic level that is transecting the stratospheric $O_3$ filament, the upper part of the jet stream and the tropospheric $H_2O$ filament. We agree that this selection is somehow arbitrary and a certain variability is likely (when we look at the results from Fig. 3-5). This may also be the case for the selection of trajectories used to analyse the vertical distribution of turbulence in Fig. 10.

However, these eight trajectories are trustworthy as they confirm the earlier identified pathways (in Fig. 3) and allow the specific aspects (Sect. 3.4) to be described in greater detail. An additional subset of trajectories for each of these locations would further extend and complicate the discussion, which we try to avoid.

5. Page 21, line 438: What is meant with "enhanced turbulence probability"?

We changed *"probability"* to *"activity"*. Generally, the turbulence metrics are only able to identify regions that likely experienced turbulence. The process itself is not resolved (see detailed discussion in Sect. 2.2).

**Technical**

Figure 1c: I assume the geopotential height contours are given in gpm. Which isolines are shown for the surface pressure? Same as in Figure 4?

We added the units for the geopotential height contours (gpm) and a contour interval for the surface pressure.

Page 6, line 163: "... but also …" => "...and also …"

Changed to *"(..) local gradients of winds (wind shear), temperature (stability) and trace gases (Kunkel et al., 2019)."*

Figure 2: The Arctic Circle is difficult to spot (but is in my opinion not too relevant).

The lines are included to indicate the midlatitudes. We changed the colour of the circles to black and thickened the lines.

Figure 3g: The figure shows max pressure, and the legend states mean pressure.

Corrected. We also corrected the description for panel (i) in the caption.

Are the figures vector graphics? When zooming in the figures (since they often contain a lot of information!), they often pixaleted and small letters were difficult to read.

Thanks for this remark and apologies for the bad quality figures. Indeed, some details were not readable. For the initial submission we included raster graphics in the word template, which led to a quality loss. We updated most figures and increased the font size wherever possible. In addition, the figures will be submitted as vector graphics and we will take care about the quality in the typeset version of the manuscript.